# Cgc-YOLO: A New Detection Model for Defect Detection of Tea Tree Seeds

**DOI:** 10.3390/s25175446

**Published:** 2025-09-02

**Authors:** Yuwen Liu, Hao Li, Kefan Yu, Hui Zhu, Binjie Zhang, Wangyu Wu, Hongbo Mu

**Affiliations:** 1College of Science, Northeast Forestry University, Harbin 150040, China; 2022213310@nefu.edu.cn (Y.L.); rttt@nefu.edu.cn (H.L.); 2022213332@nefu.edu.cn (K.Y.); 2023213237@nefu.edu.cn (H.Z.); 2022213333@nefu.edu.cn (B.Z.); 2School of Computer Science, University of Liverpool, Liverpool L69 3DR, UK; wangyu.wu@liverpool.ac.uk

**Keywords:** tea tree seed defect detection, YOLO, target detection, convolutional neural network

## Abstract

Tea tree seeds are highly sensitive to dehydration and cannot be stored for extended periods, making surface defect detection crucial for preserving their germination rate and overall quality. To address this challenge, we propose Cgc-YOLO, an enhanced YOLO-based model specifically designed to detect small-scale and complex surface defects in tea seeds. A high-resolution imaging system was employed to construct a dataset encompassing five common types of tea tree seeds, capturing diverse defect patterns. Cgc-YOLO incorporates two key improvements: (1) GhostBlock, derived from GhostNetV2, embedded in the Backbone to enhance computational efficiency and long-range feature extraction; and (2) the CPCA attention mechanism, integrated into the Neck, to improve sensitivity to local textures and boundary details, thereby boosting segmentation and localization accuracy. Experimental results demonstrate that Cgc-YOLO achieves 97.6% mAP50 and 94.9% mAP50–95, surpassing YOLO11 by 2.3% and 3.1%, respectively. Furthermore, the model retains a compact size of only 8.5 MB, delivering an excellent balance between accuracy and efficiency. This study presents a robust and lightweight solution for nondestructive detection of tea seed defects, contributing to intelligent seed screening and storage quality assurance.

## 1. Introduction

Seeds are central to crop production, human nutrition, and food security [1]. However, they face a range of challenges during storage, with tea tree seeds being particularly vulnerable. As typical recalcitrant seeds, tea seeds are not resistant to dehydration, are sensitive to low temperatures, and exhibit vigorous metabolism, making long-term preservation impossible. The primary causes of storage deterioration are fungal infections and insect infestations, which often occur simultaneously [2]. Such deterioration not only reduces seed growth potential but also impairs seedling establishment. Therefore, nondestructive automated testing solutions are being explored, as they are crucial for ensuring seed storage quality and addressing biosecurity concerns.

Traditional seed detection methods mainly rely on manual screening or recognition techniques based on conventional machine vision. Manual screening suffers from a high misjudgment rate, low recognition efficiency, and significant limitations. Early machine vision–based approaches improved efficiency to some extent but remained vulnerable to variations in lighting and background conditions. To address these issues, machine learning–based methods have been increasingly investigated. For example, Sable et al. [3] used a support vector machine (SVM) and image enhancement technology to classify soybean seeds. Similarly, Gunaseelan JayaBrindha et al. [4] integrated ant colony optimization with SVM to classify sunflower seeds, while Zhang et al. [5] employed a genetic algorithm to optimize SVM for detecting impurities in images captured by a cotton seed picking machine.

Recently, deep learning has emerged as a powerful alternative for defect detection, as it can automatically learn multi-level features from images, enabling high predictive performance in complex settings. Models such as YOLO [6,7,8] and Faster R-CNN [9] offer fast, accurate real-time detection. And the YOLO family has become widely adopted for real-time object detection tasks. With its single-stage detection and end-to-end design, it offers extremely fast processing speeds alongside strong performance in various detection tasks. Successive YOLO versions introduced improvements in Backbone architectures, multi-scale feature fusion, and detection heads. In particular, YOLO11, one of the latest models in the YOLO series, provides a refined Backbone and Neck that enhance feature extraction while reducing parameter counts. And it has shown improved performance over its predecessors in general object detection tasks. It balances accuracy, speed, and computational efficiency, which is essential for detecting small, subtle defects.

Swin Transformer–based models [10] have shown strong performance by enhancing feature extraction capabilities and overall model accuracy. Recent studies have highlighted the strong feature extraction and adaptability of the Swin Transformer. For example, Bi et al. [11] improved Swin Transformer for corn seed variety identification, while Di et al. [12] applied DETR to detect pests and seeds. Swin Transformer employs a hierarchical self-attention mechanism to capture global contextual information, enabling powerful feature representations. It performs well on complex, high-resolution images, but its global attention can dilute local feature sensitivity, which is critical for detecting small, low-contrast defects such as those on tea seeds. CNNs, by contrast, offer higher computational efficiency and faster training speeds, often achieving strong results in a short time for small and medium-sized tasks. Their local feature extraction and hierarchical convolutional structure make them particularly suitable for small-target and fine-grained defect detection while remaining deployable on resource-constrained devices, which explains why CNN-based YOLO models are widely adopted for real-time detection on resource-constrained devices. Consequently, CNNs remain the mainstream solution for mobile terminals, embedded devices, and other resource-constrained platforms. Many researchers have applied CNNs to seed classification and testing. For instance, Huang et al. [13] used Mask R-CNN to classify soybean seeds, achieving excellent results. Luan et al. [14] proposed a CNN-based method for sunflower seed variety classification; Guo et al. [15] employed a lightweight CNN to classify wheat seed varieties; and Wang et al. [16] combined a CNN with a watershed algorithm to detect maize seed defects. These works demonstrate the effectiveness of CNNs for agricultural seed detection but primarily target larger seeds or those with visually distinct defects. Detecting subtle surface flaws and defects in tea seeds, which are small, irregularly shaped, and often low-contrast, remains a significant challenge.

In the context of efficiency, lightweight object detection architectures have been developed to address computational and deployment constraints. Models such as YOLO-Nano [17], MobileNet-SSD [18], and Tiny-YOLOv4 [19] significantly reduce parameter counts and FLOPs, enabling extremely fast inference suitable for embedded devices. For example, YOLO-Nano operates with only a fraction of the parameters of YOLO11, while Tiny-YOLOv4 maintains competitive speed with modest accuracy. However, these lightweight models often struggle to detect small, low-contrast objects: reduced depth and narrower channel widths limit feature representation, and simplified Neck designs weaken multi-scale feature aggregation. In practical tea seed defect detection, such limitations can lead to missed subtle surface flaws, making the precision–recall trade-off critical. Thus, while lightweight YOLO variants are appealing for real-time applications, they are insufficient for tasks requiring high fidelity in small-target detection, which motivates the design of a more tailored architecture.

Building on these insights, YOLO11 serves as a reasonable baseline, provided its strengths are leveraged to address the deficiencies of existing lightweight models. YOLO11 refines the Backbone and Neck to enhance feature extraction while maintaining computational efficiency. Numerous YOLO variants have already been applied in seed detection tasks, highlighting the architecture’s adaptability. For instance, Kundu et al. [20] used YOLOv5 for automatic classification and quality inspection of pearl millet and corn seeds, achieving 99% accuracy and recall. Zhao et al. [21] developed the YOLO-r model for automatic detection of rice seed germination status and germination rate estimation, demonstrating robustness across complex backgrounds. Yao et al. [22] proposed a germination rate detection method for wild rice seeds based on YOLO, integrating the ECA attention mechanism, and achieving up to 98.2% germination recognition accuracy across different environments. Sun et al. [23] applied YOLOv8 to detect corn seed germination status, incorporating L-SPPF, Ghost_Detection, and RFAConv modules. The module is suitable for quickly detecting the germination state of corn seeds; Liu et al. [24] employed YOLOv9 to detect the surface defects of soybean seeds and effectively reduced the image noise by preprocessing the image. Moreover, Beyaz et al. [25] classified 84 single-bacteria and multi-bacteria beet seeds using the YOLO-tiny model of YOLOv4, which offers higher FPS performance, thereby verifying the feasibility and efficiency of deep learning in seed classification. These examples illustrate that, while YOLO architectures are highly versatile, none fully address the challenge of fine-grained defect detection in tea seeds. Tea seeds possess small, irregularly shaped, low-contrast defects, making the accurate identification of subtle surface flaws particularly challenging and reinforcing the need for improved YOLO-based architectures.

To address these challenges in tea seed quality assessment, this study focuses on surface defect detection, a task directly affecting germination rates and subsequent growth. Tea seeds are particularly prone to degradation due to their complex morphology and the small size of typical defects, making reliable detection difficult. Existing models often struggle to generalize across such visual variability. In response, we propose a novel YOLO-based framework, Cgc-YOLO, which integrates lightweight modules and attention mechanisms to improve fine-grained feature extraction and the localization of small defect regions.

The core contributions of this study are as follows:1.A dataset specifically for tea seed defect detection was developed, covering various common defect types. The dataset’s robustness was further enhanced using image augmentation techniques.2.The Backbone network was enhanced using GhostNetV2, with a specially designed Ghost_Block to capture both local and global features for small-scale targets, significantly improving model performance and computational efficiency.3.The CPCA attention module was introduced into the Neck network. Its design, combining channel priors and multi-scale spatial perception, expands the receptive field and effectively captures spatial correlations between local and global features.

The remainder of this paper is organized as follows:Section 2 describes the materials, data collection procedures, and data preprocessing, as well as the architecture of the proposed Cgc-YOLO model, with emphasis on Backbone improvements and the integration of attention mechanisms in the Neck network.Section 3 presents the experimental setup and provides a detailed analysis of the results, including comparisons with other mainstream models.Section 4 discusses the experimental findings in depth, incorporating comparisons with mainstream models and interpretability analyses.Section 5 concludes the paper by summarizing the main findings and outlining future research directions.

## 2. Methodology

### 2.1. Seed Preparation

This study used Longjing No. 43 tea tree seeds sourced from Suqian, Jiangsu, China. Longjing No. 43 was chosen not only for its wide cultivation and genetic uniformity but also because it is a high-value cultivar in the Chinese tea industry whose seeds exhibit diverse and fine-grained surface defects—such as cracks, bare areas, and mold growth—that are visually subtle and challenging to detect. These characteristics make it an ideal benchmark for evaluating small-defect detection models. Furthermore, its consistent morphology across production regions facilitates dataset standardization and cross-study comparability. Seeds were cleaned, impurities removed, and pests eliminated through short-term refrigeration. To prepare moldy samples, seeds were briefly hydrated and stored under warm, humid conditions until visible fungal growth appeared.

All prepared tea seeds were divided into five categories: cracked seeds, moldy seeds, bare seeds, large improved seeds, and small improved seeds. Representative images of each seed type are shown in Figure 1.

### 2.2. Image Acquisition

The image acquisition system consisted of two shooting devices: a high-resolution CMOS industrial camera (Hikon MV-CS060-10UM-PRO, China) with a resolution of 3072 × 2048 pixels and a mobile phone (Honor Magic VS2) with a resolution of 3072 × 4096 pixels. This dual-device setup introduced cross-device variability to improve model generalizability. A highly uniform, adjustable light source and a diffuse light guide plate were used as the shooting background to minimize mirror reflections that could obscure subtle defects.

Seeds were placed centrally, and the camera was fixed at an optimal shooting distance of approximately 12–15 cm. Key exposure parameters were kept constant to ensure consistency. The setup allowed repositioning of seeds to capture images from multiple orientations. It was also designed to allow three-way adjustments in height, brightness, and angle to comprehensively record the surface characteristics of seeds, as illustrated in Figure 2.

### 2.3. Image Processing and Enhancement

To ensure input consistency and model compatibility, the original image resolution was adjusted to 640 × 640 pixels. Each tea tree seed was then manually annotated with a rectangular bounding box using the LabelMe labeling tool, with seeds classified into five categories: cracked seeds, moldy seeds, bare seeds, large improved seeds, and small improved seeds. After annotation, the corresponding JSON files were generated.

To improve the performance and robustness of the model, data enhancement techniques are applied to the images in the dataset. In this study, Albumentations is used for data enhancement. The chosen augmentations were tailored to simulate realistic variations in tea seed images, thereby increasing dataset diversity and reducing overfitting.

To enhance model performance and robustness, data augmentation techniques were applied to the dataset using the Albumentations library in Python [26]. These augmentations were designed to address overfitting and improve dataset diversity.

A combination of augmentation methods was employed. VerticalFlip, HorizontalFlip, and Transpose transformations were each applied with a probability of 0.6 to simulate the arbitrary orientations in which seeds may appear during handling, storage, or image capture. Random rotation within 0–360° (probability 0.5) further accounted for unrestricted seed placement, a condition frequently observed in practical inspection scenarios. Grid Distortion (border_mode = 4, interpolation = 1) divided images into grids and applied subtle warping to mimic minor distortions caused by lens curvature or uneven positioning on the imaging surface, ensuring that the model remains robust to geometric imperfections.

Two types of noise were introduced. Gaussian noise with mean 0 and variance 10–50 replicated image degradation in low-light conditions or when using low-cost imaging devices, with the variance range chosen to avoid obscuring key seed features while still representing realistic noise levels. ISO noise adjustments (hue variance 0.01–0.05; multiplier 0.1–0.5) simulated natural discoloration from seed defects and variations in illumination or surface reflectance, ensuring the model could generalize to different lighting conditions.

In practical applications, weather changes can affect the model’s recognition performance. To address this, we employed the weather data augmentation techniques available in the Albumentations library to simulate environmental variations caused by different weather conditions. Weather-based augmentation was implemented via Random_rain, with raindrop size set to 1.0 and brightness_coefficient fixed at 0.7. These parameters were tuned to create realistic rainfall effects without fully obscuring surface features, representing potential outdoor or unshielded imaging environments during rainy or overcast days.

By introducing these augmentation techniques, the dataset incorporates subtle variations in illumination, texture, and background uniformity that are commonly encountered in industrial seed inspection. These variations compel the model to focus on defect-specific features—such as cracks, discoloration, or mold growth—rather than relying on consistent lighting or background smoothness. As a result, the augmented dataset not only reflects the variability of real-world seed imagery but also strengthens the model’s ability to detect fine surface defects under diverse imaging conditions.

After applying the above data augmentation process, the final dataset comprised 6479 tea tree seed images. Using stratified random sampling, the dataset was divided into training, validation, and testing sets in a ratio of 7:2:1, ensuring balanced category representation across all subsets. Specifically, the training set contained 4535 images, the validation set contained 1295 images, and the testing set contained 649 images.

### 2.4. Tea Tree Seed Detection and Identification Model

#### 2.4.1. YOLO11

YOLO [6] is an end-to-end object detection framework that processes images through three main components: the Backbone network for feature extraction, the Neck network for multi-scale feature fusion, and the head network for final target localization and classification. While earlier YOLO versions have achieved a strong balance between speed and accuracy, they face limitations in capturing both fine-grained local features and broader contextual information simultaneously.

YOLO11 addresses these challenges with a redesigned Backbone and enhanced attention mechanisms, offering notable improvements in recognition accuracy, model compactness, and inference speed. As illustrated in Figure 3, its Backbone integrates C3k2, a customizable convolution block that flexibly adjusts kernel sizes to expand the receptive field and balance local–global feature perception. This enables the network to better detect small defects while maintaining context awareness for larger structures. Complementing this, the C2PSA module introduces position-sensitive attention, refining the model’s ability to capture spatial relationships, contours, and local structural details essential for distinguishing subtle defects. Together, these innovations enhance YOLO11’s adaptability to complex visual patterns in seed imagery, as summarized in Figure 3.

YOLO11 offers excellent real-time detection and high-precision identification capabilities, making it suitable for various target detection tasks. However, it still faces limitations when detecting irregularly shaped defects in tea seeds. Its rectangular bounding box approach struggles to accurately describe complex edges, often resulting in incomplete detection or boundary deviations. Furthermore, when deployed on embedded devices with limited computational resources, its real-time performance and efficiency can be constrained.

As shown in Figure 4, we introduced two key improvements to YOLO11:(1)The original C3k2 module in the YOLO11 Backbone was replaced with a Ghost_Block module. Ghost_Block uses cost-effective linear operations to generate additional feature maps from intrinsic feature representations, significantly reducing computational overhead. This replacement not only improves model efficiency but also enhances its ability to capture long-range dependencies across feature dimensions, allowing for a more accurate representation of the global shape and texture of irregular tea seed defects.(2)The Neck network was upgraded with the Cross-Path Channel Attention (CPCA) module, which strengthens boundary feature representation through adaptive attention across both spatial and channel dimensions. CPCA selectively emphasizes relevant edge features, increasing the model’s sensitivity to subtle contour variations and improving detection of fine, irregular surface defects often missed by conventional object detectors.

Together, these architectural enhancements make Cgc-YOLO better suited for real-world scenarios involving complex morphological patterns, while ensuring low-latency inference on embedded platforms.

**Figure 4 sensors-25-05446-f004:**
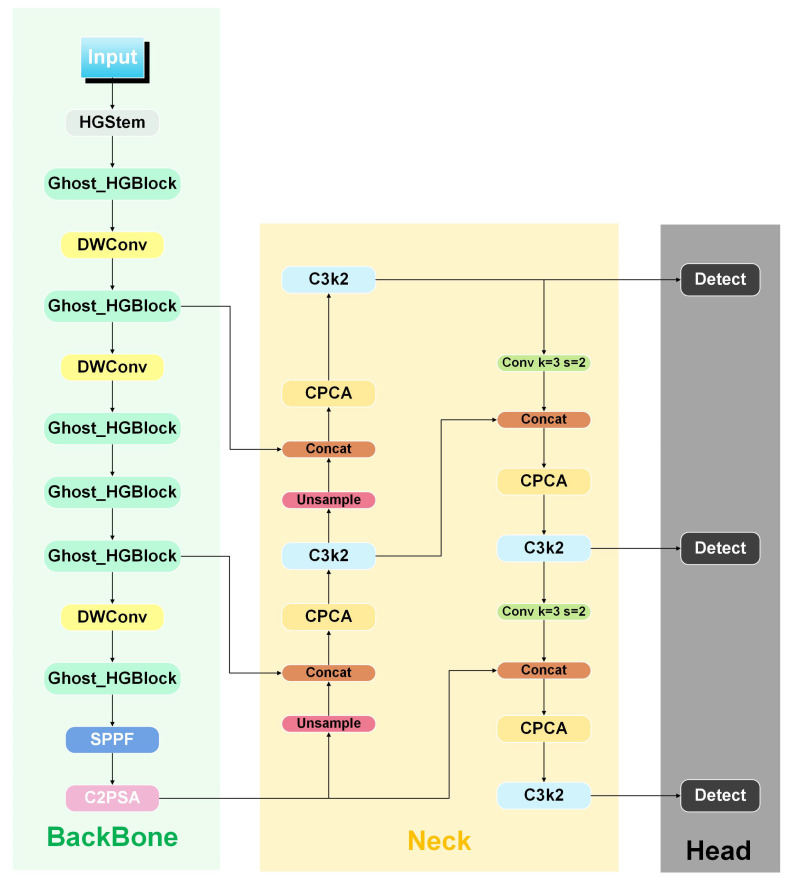
Model Architecture of Cgc-YOLO.

#### 2.4.2. Lightweight Backbone Network GhostNetV2

The Backbone of the Cgc-YOLO network adopts the efficient and lightweight GhostNetV2 architecture, which is specifically optimized for environments with limited computing resources. Its core components include an enhanced Ghost module and depthwise separable convolution, both designed to improve feature extraction capability. This makes it suitable for tasks such as tea seed detection, which requires high-precision recognition under constrained computational conditions. As shown in Figure 4, the Backbone uses the GhostNetV2 architecture combined with a multi-layer DWConv module to construct a feature extraction path. The Ghost_HGBlock module generates the main features, while DWConv provides efficient convolutional support. This overall design improves inference speed while maintaining strong multi-scale feature representation capabilities.

As a lightweight feature extraction model, GhostNet [27] is composed of two stacked Ghost modules. As illustrated in Figure 5, the basic working principle of the Ghost module is as follows: First, the primary features are extracted from the input features X RH × W × C through a 1 × 1 convolution; next, these primary features are linearly transformed using depthwise separable convolution, thereby enriching the information contained in the feature maps; finally, the two sets of features are concatenated. The main computational process of the Ghost module can be expressed as(1)$$Y^{\prime }=X*F_{1\times 1}$$(2)$$Y=\mathrm{Concat}(\left[Y^{\prime },Y^{\prime }*F_{\mathrm{dp}}\right])$$
where denotes the convolution operation, represents point-by-point convolution, and intrinsic features refer to features whose size is typically smaller than the original output features. $F_{dp}$ is the depthwise convolutional filter, and Y denotes the output feature. The Ghost module significantly reduces computational costs by generating spatial information through 3 × 3 depthwise convolution, while the remaining features are produced via 1 × 1 pointwise convolution.

The GhostV2 bottleNeck module incorporates the DFC attention mechanism. As illustrated in Figure 6, the output features of the Ghost module are enhanced by decoupling the fully connected (FC) layer with fixed weights into two FC layers, applied separately in the horizontal and vertical directions. As shown in Figure 5, this approach enables the model to capture long-range dependencies between pixels across different dimensions. The specific process, detailed below, effectively reduces the computational complexity of the attention module.(3)$$a_{\mathrm{hw}}=\sum_{h^{\prime },w^{\prime }}F_{\mathrm{hw},h^{\prime }w^{\prime }}*Z_{h^{\prime }w^{\prime }}$$
where * denotes the symbol of the multiplication operation, $a_{hw}$ represents the output features after vertical and horizontal transformations, $F_{hw}$ and $F_{h^{\prime }w^{\prime }}$ represent the learnable weights in the horizontal and vertical directions, and $Z_{h^{\prime }w^{\prime }}$ denotes the features in both horizontal and vertical directions.

The Ghost_HGBlock employs multiple GhostV2 bottleNeck layers in series to progressively extract features. The output from each layer is transmitted to a concatenation layer for feature fusion via skip connections. Subsequently, the SC (Spatial Convolution) and EC (Efficient Channel Attention) modules are introduced to strengthen inter-channel dependencies and improve feature representation. Finally, the fused features are combined with the input through a residual connection, completing the construction of the Ghost_HGBlock.

#### 2.4.3. Channel Prior Convolutional Attention

CPCA (Channel Prior Convolutional Attention) [28] is a lightweight and efficient attention mechanism operating in the channel space. It employs multi-scale depthwise separable convolution to dynamically distribute attention across both channels and spatial dimensions, thereby enhancing the focus on key regions. Since seed defects often contain rich textures and detailed features, spatial attention plays a crucial role in accurately identifying subtle changes within specific areas. Moreover, certain complex seed structures, such as moldy seeds or bare seeds, require semantic detection of colors and edges, making the channel attention mechanism essential for dynamically weighting different channels. The architecture of the CPCA module is illustrated in Figure 5.

As shown in Figure 5, CPCA comprises channel attention (CA) and spatial attention (SA). By employing a multi-scale depthwise separable convolution module, it effectively captures spatial relationships while preserving existing channels. Unlike CBAM (Convolutional Block Attention Module), CPCA leverages a depthwise convolution module to establish spatial attention elements. The depth convolution module uses extended convolution kernels with different sizes to capture the spatial relationship between pixels. By employing multiple scales, CPCA can extract key information while minimizing computational overhead.

For the input feature map F, average pooling and max pooling are first applied, aggregating spatial information from two independent perspectives to generate two distinct spatial descriptors. This dual-pooling strategy is based on the assumption that defect-relevant features in tea seeds exhibit distinguishable patterns in either average or maximum intensity distributions. Moldy regions, for example, show consistent color shifts captured by average pooling, while cracks or bare spots display local intensity peaks captured by max pooling, avoiding over-reliance on a single statistical measure. These descriptors are then passed through a shared multi-layer perceptron (MLP), and the result generates the channel attention map CA(F), which accurately reflects the true feature distribution and enhances the network’s segmentation performance. The channel prior is obtained by performing element-wise multiplication between the input feature map F and the channel attention map CA(F), yielding an effectively weighted channel prior. Multi-scale spatial information is then extracted by feeding this result into the depth-separable convolution module with kernel sizes 5 × 5, 1 × 7, 1 × 11, 1 × 21, 7 × 1, 11 × 1, 21 × 1—vital for detecting defects of varying sizes from tiny spots to large cracks. After feature fusion, a 1 × 1 convolution is applied for channel fusion, generating the spatial attention map SA(F). Depthwise convolutions here apply each kernel to individual channels, reducing computation compared with standard convolutions in CBAM. The calculation of channel attention and spatial attention can be expressed as follows: (4)CA(F)=Sigmoid(MLP(AvgPool(F))+MLP(MaxPool(F))(5)$$\mathrm{SA}\left(F\right)={\mathrm{Conv}}_{1x1}(\sum_{i=0}^{3}{\mathrm{Branch}}_{i}\left(\mathrm{DwConv}\left(F\right)\right)$$
where DwConv denotes the depth convolution, ${\mathrm{Branch}}_{i}$ denotes the i-th branch, and each ${\mathrm{Branch}}_{i}$ is a constant connection.

In contrast to CBAM, which applies standard convolutions in spatial attention, CPCA adopts depthwise separable convolutions that process each channel independently with dedicated kernels. This design reduces redundant parameter learning: for seed feature maps, the parameter count of CPCA is significantly lower than CBAM. The key reason is that depthwise convolutions in CPCA scale with the number of channels and kernel size, while CBAM’s standard convolutions involve cross-channel parameter interactions. This parameter reduction, combined with shared MLP in channel attention that avoids redundant learning, results in a lower computational load. Such efficiency is critical for deploying the detection system on seed sorting equipment with limited on-board computing resources, ensuring real-time processing of high-resolution seed images.

Overall, CPCA applies channel attention followed by spatial attention. Channel attention first aggregates spatial information via average and max pooling, then processes it through a shared multi-layer perceptron (MLP) to produce the channel attention map. Element-wise multiplication between the input features and the channel attention map produces the refined feature, which is subsequently passed through the depthwise separable convolution module to generate the spatial attention map. Channel fusion is then applied, and the spatial attention map is multiplied element-wise with the channel attention map to produce the final output feature. This integration enhances feature representation, with the calculation process defined as follows: (6)$$F_{c}=\mathrm{CA}\left(F\right)⊗F$$(7)$$\hat{F}=\mathrm{SA}\left(F_{c}\right)⊗F_{c}$$
where $F_{c}$ denotes the refined feature, $\hat{F}$ denotes the final output feature.

## 3. Results and Discussion

### 3.1. Experimental Configuration and Evaluation Indicator

The hyperparameters used for training the model are listed in Table 1, while the specifications of the training equipment are provided in Table 2. All model construction, training, and testing were carried out using Visual Studio Code. The parameters and experimental environment configuration are summarized in the tables below.

To evaluate the effectiveness of the proposed tea tree seed detection model, this study employs Box Precision (P), Recall (R), Average Precision (AP), and mean Average Precision (mAP) as performance metrics. Model complexity is assessed based on the number of parameters, while detection speed is evaluated using the computational cost, measured in FLOPs. The calculation formulas for these metrics are as follows:(8)$$\mathrm{Precsion}=\frac{\mathrm{TP}}{\mathrm{TP}+\mathrm{FP}}\times 100\%$$(9)$$\mathrm{Recall}=\frac{\mathrm{TP}}{\mathrm{TP}+\mathrm{FN}}\times 100\%$$(10)$$\mathrm{AP}=\int_{0}^{1}P(R)\mathrm{dR}$$(11)$$\mathrm{mAP}=\frac{\sum_{i=1}^{cAP_{i}}}{C}\times 100\%$$

In the formulas, TP is true positives (correctly identified defective samples), FP is false positives (non-defective samples mislabeled as defective), and FN is false negatives (missed defective samples). AP balances precision and recall, and mAP is the average AP over all k categories.

In addition to the evaluation metrics, the distribution characteristics of the dataset play a crucial role in validating the model’s generalization ability. The distribution of these images is illustrated in Figure 7, which provides insights into the data composition and category balance of the constructed tea seed defect dataset.

### 3.2. Comparative Experiment of CPCA Module: Impact on Detection Performance and Efficiency

To further investigate the optimal deployment of the CPCA module within the network structure and its specific contributions to detection performance, three deployment modes were designed: placing CPCA in the Backbone, in the Neck, and in both the Backbone and Neck simultaneously. All experiments were conducted while keeping the remaining network structures and hyperparameter settings consistent to ensure fairness and validity in the comparison.

YOLO11+CPCA-Backbone embeds the CPCA module into several key layers of the Backbone to improve channel modeling between low- and mid-level features, thereby enhancing the representation capability for small targets. YOLO11+CPCA-Neck integrates the CPCA module into the feature fusion path to optimize the fusion and representation of high-level semantic features. YOLO11+CPCA-Backbone&Neck combines both approaches to establish a stronger global feature interaction mechanism. The results are presented in Table 3 and Figure 8.

CPCA-Backbone shows some effectiveness in enhancing low-level feature representation, with a 0.011 improvement in mAP50 from 0.953 to 0.964 and the smallest parameter increases only 0.13M more than the base model. However, its FPS value of 399.52 is relatively high, which makes it less suitable in scenarios requiring a balanced performance across multiple metrics, as it may not meet the practical demand for a harmonious trade-off between accuracy and other efficiency-related factors. And CPCA-Neck strengthens high-level semantic representation, as mAP@50-95 increased from 0.918 to 0.935 by 0.017, making it applicable to scenarios like real-time video analysis that require a balance between accuracy and efficiency. Although CPCA-Backbone&Neck was expected to enhance the interaction between low- and high-level features simultaneously, experimental results show its performance gains matching CPCA-Neck’s mAP@50-95 of 0.935 do not justify the increased model complexity (same GFLOPs and parameters as CPCA-Neck but lower FPS). Overall, deploying the CPCA module in the Neck achieves an optimal balance, leveraging channel modeling and feature fusion to obtain a 0.017 improvement in mAP@50-95 with a reasonable trade-off between accuracy and computational efficiency, which is favorable for practical applications needing reliable and efficient object detection.

### 3.3. Ablation Experiments

To verify the effectiveness of the proposed module, a series of ablation experiments was conducted on the tea seed dataset. During these experiments, the model’s hyperparameters remained unchanged. Several comparative setups were designed by progressively integrating the optimization method into the original network. The improved models were evaluated using the metrics described in the previous section, and the results are presented in Table 4. The 95% confidence intervals (CIs) for all performance metrics were calculated using the following formula: (12)$$\mathrm{CI}=\bar{x}\pm t_{\alpha /2,\;n-1}\times \frac{s}{\sqrt{n}}$$
where $\bar{x}$ denotes the mean value of the metric across repeated trials, *s* represents the sample standard deviation, *n* is the number of independent experimental trials, and $t_{\alpha /2,\;n-1}$ is the critical value of the t-distribution with n−1 degrees of freedom at the 95% confidence level (α=0.05).

The experimental results indicate that, compared with the original YOLO11 model, the improvements achieved by the CPCA variants (CPCA-Neck and CPCA-Backbone) are substantial. Specifically, YOLO11+CPCA-Neck shows increases of 1.8%, 1.0%, and 1.7% in Box(P), mAP50, and mAP50-95, respectively, with a notable enhancement in detection efficiency. The CPCA module strengthens the model’s focus on key regions and feature dimensions by jointly modeling channel and spatial attention.

In addition, incorporating the GhostBlock further boosts performance, with YOLO11+GhostHGNetV2 achieving 97.5% mAP50, representing gains of 2.2% and 3.6% over the baseline. Among all enhancements, Cgc-YOLO delivers the most comprehensive performance gains, achieving Box(P) of 94.3%, R of 90.2%, mAP50 of 97.6%, and mAP50-95 of 94.9%, corresponding to improvements of 4.4%, 2.0%, 2.3%, and 3.1% over YOLO11, respectively. These results demonstrate that the integrated use of the CPCA and GhostBlock modules optimizes both the Backbone and Neck, significantly improving detection efficiency and overall model performance.

### 3.4. Comparative Experiments

In the comparative experiment stage, we conducted an in-depth comparison between Cgc-YOLO and several mainstream models in the field of object detection. For fair comparison, all YOLO models (baselines and Cgc-YOLO) were retrained under unified conditions: dataset split as 7:2:1 (training/validation/test); standardized parameters including batch size 8300 epochs, initial learning rate 0.01, momentum 0.937, and weight decay 0.0005. All experiments ran on a Windows 11 system with a NVIDIA GeForce RTX 4060 Laptop GPU, using Python 3.12, PyTorch 2.6.0, and CUDA 12.7. This consistency ensures performance differences stem from architectural improvements.

When compared with several widely used YOLO series models, including YOLOv3, YOLOv5, YOLOv6, YOLOv8, YOLOv9, YOLOv10, and YOLO11, the results demonstrated that Cgc-YOLO achieved superior detection accuracy. Specifically, the Box(P) value reached 94.3%, and its mAP50 was 1.8%, 1.5%, and 2.5% higher than those of YOLOv5, YOLOv8, and YOLO11, respectively. Likewise, mAP50–95 improved by 2.0%, 2.3%, and 3.1% compared with YOLOv5, YOLOv10, and YOLO11, respectively. This improvement may be attributed to the more effective integration of high-level semantic information with low-level texture details in Cgc-YOLO, which other models may not fully exploit. Compared with YOLOv5, YOLOv6, YOLOv8, YOLOv10, and YOLO11, Cgc-YOLO achieved R-value increases of 5.1%, 3.5%, 4.1%, 3.2%, and 4.4%, respectively. This enhancement can be linked to the inclusion of the small-target enhancement module, which enables the model to extract more effective information at smaller scales, an area where some competing models struggle. Notably, only YOLOv3 exhibited performance comparable to Cgc-YOLO; however, the model size of Cgc-YOLO is just 8.5 MB, only 4.3% of YOLOv3’s size. This demonstrates that Cgc-YOLO achieves remarkable lightweight efficiency while maintaining strong performance. This compactness is primarily due to the adoption of GhostNetV2, which enhances feature expression efficiency while reducing redundant computations.

These results demonstrate that Cgc-YOLO strikes an effective balance between accuracy and lightweight design, showing strong potential for practical use in industrial production. The performance comparison of different models is presented in Figure 9 and Table 5. Moreover, except for YOLOv3, the paired *t*-test results between Cgc-YOLO and the other six models (YOLOv5, YOLOv6, YOLOv8, YOLOv10, YOLO11, and YOLOv12) were all less than 0.005 (10 trials), indicating that the performance gains are statistically significant rather than incidental. And comparisons between Cgc-YOLO and YOLOv5, YOLOv6, YOLOv8, YOLOv10, YOLO11, and YOLOv12 all exhibit large effect sizes (Cohen’s d > 0.8), with only a small effect size observed in comparison to YOLOv3 (d = 0.24). After Bonferroni correction (total number of tests k=7, corrected $\alpha_{\mathrm{corrected}}\approx 0.0071$), the differences in mAP@50 between Cgc-YOLO and YOLOv5, YOLOv6, YOLOv8, YOLOv10, YOLO11, and YOLOv12 were all statistically significant (*p* < 0.001), indicating that its performance improvements hold substantial practical value.

The precision–recall (P–R) curve is commonly used to evaluate model performance in defect classification, as it reflects the average precision (AP) for each defect category and illustrates the relationship between precision and recall. The area under the P–R curve (AUC–PR) serves as a key metric for assessing overall performance; higher values indicate that the model can maintain both high recall and high precision. As shown in Figure 10, Cgc-YOLO achieves higher average precision and a more balanced P–R curve across all defect types, indicating a more stable and uniform detection capability across categories. In contrast, some YOLO series models perform well in certain categories but exhibit noticeable fluctuations in others, limiting overall performance. For example, YOLOv3 excels in detecting moldy defects but shows low recall for large-scale improved varieties, highlighting its weakness in fine-grained feature modeling. Cgc-YOLO, by comparison, demonstrates consistent improvements across the three major defect categories: Compared with YOLO11, Cgc-YOLO improves crack detection accuracy by 1.4%, bare-part detection by 4.1%, and moldy-part detection by 5.3%. This demonstrates that Cgc-YOLO not only enhances overall detection accuracy but also effectively mitigates performance imbalances across categories. This improvement stems from the CPCA module’s ability to facilitate semantic interaction between channel and spatial dimensions, enabling the model to capture multi-scale, fine-grained defect features more comprehensively. As a result, its generalization ability and robustness are improved. The P–R curves for the different models are shown in Figure 10.

### 3.5. Visualization

Figure 11 presents a comparison of detection results, with confidence scores displayed next to each detection box. The experimental results demonstrate that Cgc-YOLO significantly outperforms other models in detecting surface defects of tea seeds. For moldy seeds, Cgc-YOLO achieves a confidence score of 0.95, compared with 0.89 for the original YOLO11 model, an increase of 0.06. For cracked seeds, the confidence score is 0.95, which is 0.03 higher than the 0.92 score of YOLO11. Overall, Cgc-YOLO shows a clear improvement in detection confidence compared with the original model.

Grad-CAM is an interpretable visualization method for neural networks that generates heatmaps using the model’s gradient information, highlighting key image regions influencing specific category predictions. Building on prior work, such as Umair et al.’s use of Grad-CAM to visualize discriminative regions [29] and Yu et al.’s application in spectral defect analysis for fruit classification [30], we employ Grad-CAM to enhance the interpretability of our model. In this study, we performed a comprehensive Grad-CAM analysis across multiple representative layers of both YOLO11 and Cgc-YOLO to examine their attention distribution and defect localization behavior. This extended approach, which goes beyond analyzing a single feature map, offers a deeper understanding of the models’ internal focus and decision-making processes, including how they learn to prioritize features and the problem of model false detection and omissions.

As illustrated in Figure 12, during the detection of surface defects on tea seeds, the heatmaps generated by Cgc-YOLO show more intense and concentrated activation in the defect regions. This pattern directly reflects the model’s functional learning: the CPCA attention mechanism and GhostBlock modules enable it to learn to weight defect-related features more heavily, strengthening the encoding of critical defect characteristics, e.g., edge contours and texture anomalies in intermediate layers. Consequently, the concentrated heatmaps indicate that Cgc-YOLO has effectively learned to associate these regions with defect categories, reducing the risk of misclassifying background noise as defects—a common source of decision errors in less focused models. In contrast, YOLO11 displays lighter activation around defect areas, weaker responses to key features, and higher sensitivity to background noise. This scattered attention reveals limitations in its functional learning: without targeted feature enhancement, it fails to robustly distinguish defect-specific patterns from irrelevant background information, e.g., natural surface variations of tea seeds. The resulting heatmaps suggest that YOLO11 sometimes allocates gradient weight to non-defect regions, leading to decision errors such as missed detections when defect activation is too weak or false positives when background noise triggers undue attention.

These qualitative findings align with the quantitative results reported in Table 5, where Cgc-YOLO achieves higher precision and recall rates. The concentrated activation in defect regions confirms that its functional learning prioritizes meaningful defect features, directly reducing false negatives; meanwhile, the reduced sensitivity to background noise minimizes false positives. In contrast, YOLO11’s scattered attention correlates with its lower precision, as decision errors stem from its failure to learn a clear boundary between defect and non-defect regions. Overall, Cgc-YOLO’s superior performance arises from functional learning focused on defect-relevant regions, and the improved feature extraction and spatial perception capabilities of Cgc-YOLO are mainly attributed to the integration of the CPCA attention mechanism and GhostBlock modules, which enhance the model’s ability to capture both local and global contextual information.

## 4. Discussion

### 4.1. Result Interpretation

The experimental results demonstrate that the proposed Cgc-YOLO model significantly improves detection accuracy and robustness in identifying tea tree seed defects. Specifically, Cgc-YOLO achieves an mAP50 of 97.6% and an mAP50-95 of 94.9% on the constructed tea seed defect dataset, markedly outperforming the baseline YOLO11 model. Compared with YOLO11, Cgc-YOLO improves mAP50 by 2.3% and mAP50-95 by 3.1%, indicating superior precision in both coarse and fine localization tasks. This performance gain primarily stems from integrating the GhostNetV2 Backbone, which enhances computational efficiency and strengthens long-range feature extraction, along with the CPCA attention mechanism in the Neck network, which improves perception of object boundaries and local texture details. These enhancements enable Cgc-YOLO to better capture the diverse and irregular shapes of tea seed defects that traditional rectangular detectors struggle to detect.

Additionally, the lightweight design of Cgc-YOLO is reflected in its compact model size of only 8.5 MB, facilitating deployment on resource-constrained embedded devices without compromising accuracy. Overall, these results demonstrate that Cgc-YOLO achieves an effective balance between detection accuracy and computational efficiency, making it well-suited for real-world agricultural applications requiring fast and precise defect detection.

### 4.2. Limitation

In this study, the proposed Cgc-YOLO model demonstrated excellent performance in detecting surface defects in tea tree seeds. However, several limitations remain to be addressed in future work. First, the model is specifically designed and optimized for the unique visual characteristics of tea tree seeds, such as irregular textures and subtle defects. Its effectiveness on other agricultural products or industrial defect detection tasks remains uncertain, and thus, its generalizability requires further investigation.

Second, the dataset was primarily collected in controlled laboratory settings with consistent lighting and backgrounds. In real-world applications, varying lighting conditions, background interference, and seed occlusion may impact detection accuracy. Consequently, the current results might not fully capture the model’s robustness under more complex and diverse scenarios. Future research should explore applying the model to other seed types or agricultural products with similar defect features and validate its performance across varying environmental conditions, including different lighting, occlusion, and imaging devices.

Third, the model exhibits non-negligible false detection issues in practical testing, particularly in two scenarios. As shown in Figure 13, when seeds have natural texture variations resembling defects, such as uneven coloration caused by inherent seed surface patterns. The model incorrectly labels these normal features as defects. Additionally, the model occasionally misclassifies background noise or seed edges as defects. These issues arise partly because the training dataset has limited samples of normal seeds with complex natural textures and insufficient annotations for overlapping scenarios, leading the model to overfit to partial defect features.

Fourth, although the model’s lightweight design enables deployment on embedded devices, there is a trade-off between detection speed and accuracy in high-throughput scenarios. When processing batches of seeds with high density or overlapping arrangements, the inference speed decreases compared with single seed detection, which may affect the efficiency of large-scale seed screening in industrial production lines.

### 4.3. Broader Impact

This study confirms the potential of lightweight YOLO for agricultural small-target defect detection. On one hand, the compact 8.5 MB model size overcomes deployment constraints on embedded devices, such as the Jetson Nano edge computing terminal, meeting real-time detection requirements. On the other hand, CPCA’s ability to accurately capture local defects provides valuable insights for other fine-grained visual tasks, such as detecting mold on medicinal materials or identifying fruit surface damage. Future integration of multimodal data, e.g., combining near-infrared spectral features, could further enhance early warning capabilities for defect detection, contributing to the advancement of refined management in smart agriculture.

## 5. Conclusions

Detection of surface defects in tea tree seeds is crucial for seed storage quality. To address the challenges posed by the variety, complexity, and multi-scale features of tea seed defects, this study proposed Cgc-YOLO, an improved detection model based on YOLO11. Key enhancements include replacing the Backbone with GhostNetV2, incorporating the GhostBlock module, and integrating the CPCA attention mechanism, which effectively captures spatial relationships.

Using a high-resolution imaging system, a dataset was constructed from five common tea seed types. Experimental results showed that Cgc-YOLO achieved an mAP50 of 97.6% and mAP50-95 of 94.9%, improving over YOLO11 by 2.3% and 3.1%, respectively. With a lightweight model size of only 8.5 MB, Cgc-YOLO outperformed mainstream detection models, particularly in identifying moldy and bare seeds. Visualization analyses confirmed that Cgc-YOLO better focuses on key defect features and target areas, effectively handling multi-type, complex, and multi-scale defects.

For tea seed production bases, YOLO-based detection reduces labor costs and improves screening accuracy, but critical limitations remain. The dataset, though covering five seed types, is limited to one regional base, lacking diversity in cultivation, storage, and origins, restricting adaptability across regions. High test performance is tempered by similar training-test data distribution (same sample batch), raising overfitting risks for unseen samples. The model is unvalidated in real production environments with variable lighting/occlusions and lacks multi-site comparisons with third-party data or manual benchmarks. Additionally, detecting tiny defects (under 5 pixels) is hindered by insufficient low-resolution feature information. Our future work will focus on expanding the research scope to enhance the practical applicability of the detection system. The dataset will be expanded by collecting samples from multiple major tea-producing regions across different latitudes, covering more tea seed varieties and including seeds stored under diverse environmental conditions. Rare defect types will be augmented to enrich the diversity of defect samples. A dedicated small-target detection head will be integrated into the model architecture to enhance the recognition of tiny defects. Field validation will be conducted in collaboration with tea seed production bases, using mobile imaging platforms to collect real-world data under natural environmental conditions. Meanwhile, domain adaptation algorithms and synthetic data generation techniques will be explored to further boost the model’s robustness across different application scenarios.

In summary, Cgc-YOLO represents a significant advancement, offering strong technical support for the intelligent detection of tea tree seed defects and laying a solid foundation for the development of related detection equipment.

## Figures and Tables

**Figure 1 sensors-25-05446-f001:**
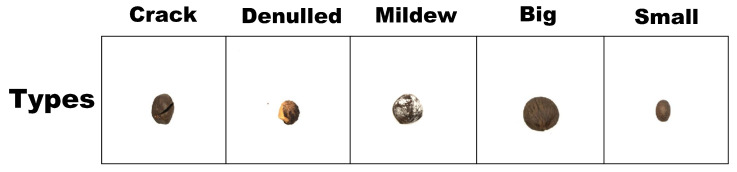
Five different tea tree seed type markers, including defective and good seeds: cracked seeds; denulled seeds; moldy seeds; large good seeds; small good seeds.

**Figure 2 sensors-25-05446-f002:**
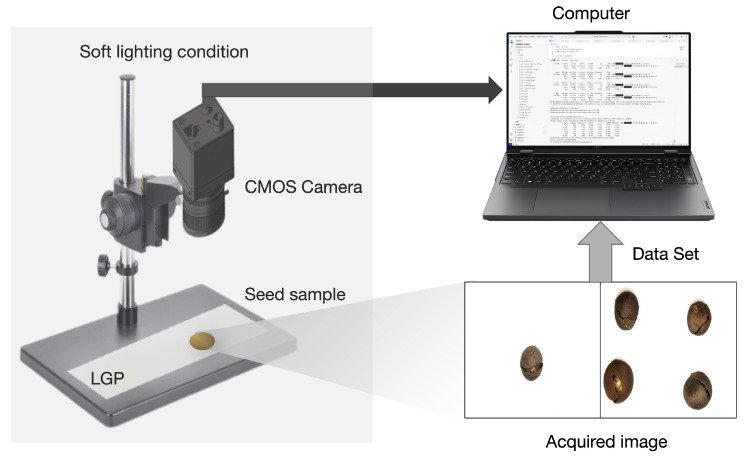
Image Acquisition Process of the Shooting System.

**Figure 3 sensors-25-05446-f003:**
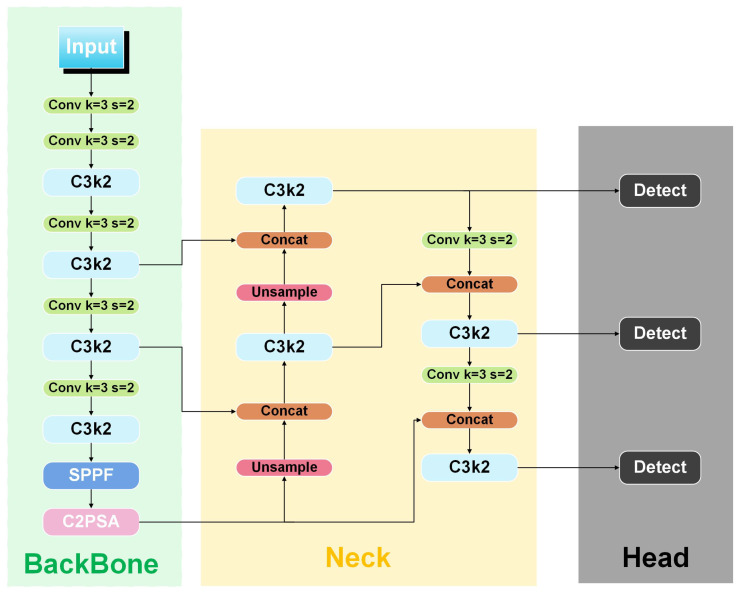
Model architecture of YOLO11.

**Figure 5 sensors-25-05446-f005:**
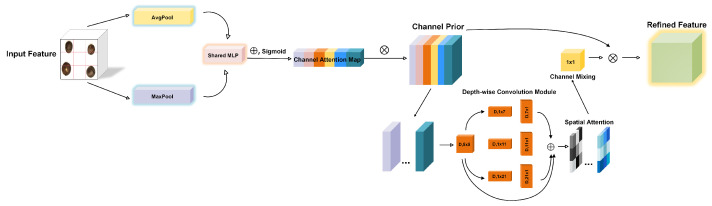
CPCA features an overall structure comprising sequential placement of CA and SA. Spatial information of the feature maps is aggregated by the CA through operations such as average pooling and max pooling. The spatial information is subsequently processed through a shared MLP and added to produce the CA map. The CA is obtained by element-wise multiplication of the input feature and the CA map. Subsequently, the CA is input into the depth-wise convolution module to generate the SA map. The convolutional module receives the SA map for channel mixing. Ultimately, the refined features are obtained as the output by element-wise multiplication of the channel mixing result and the channel prior.

**Figure 6 sensors-25-05446-f006:**
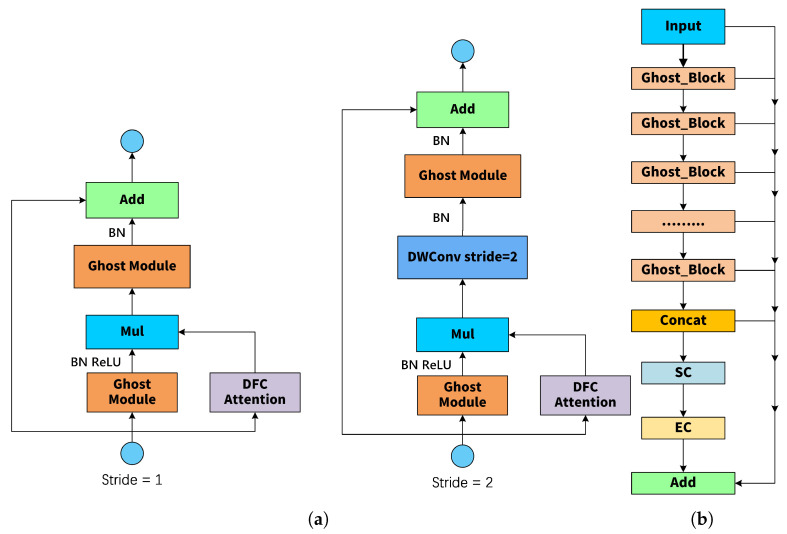
(**a**) Architecture of the Ghost_Block module; (**b**) Architecture of the Ghost_HGBlock module.

**Figure 7 sensors-25-05446-f007:**
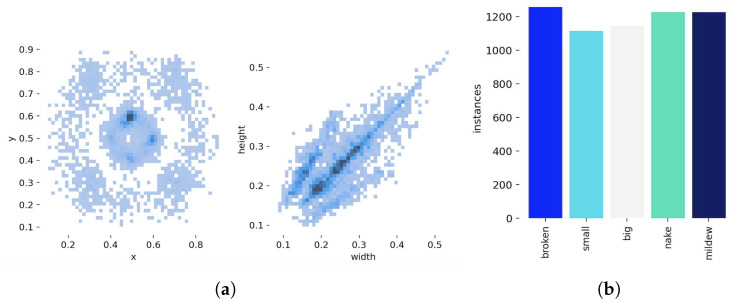
(**a**) The feature distribution of image data; (**b**) the distribution of images of different types.

**Figure 8 sensors-25-05446-f008:**
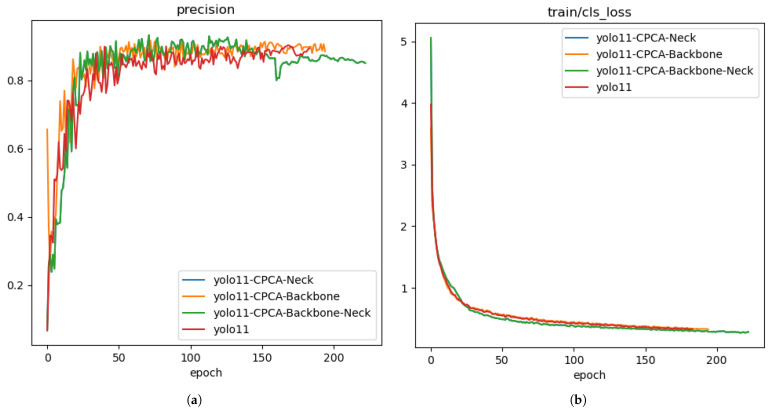
The performance of different models in the comparative experiment of CPCA module: (**a**) The precision curve; (**b**) The classification loss on the train set.

**Figure 9 sensors-25-05446-f009:**
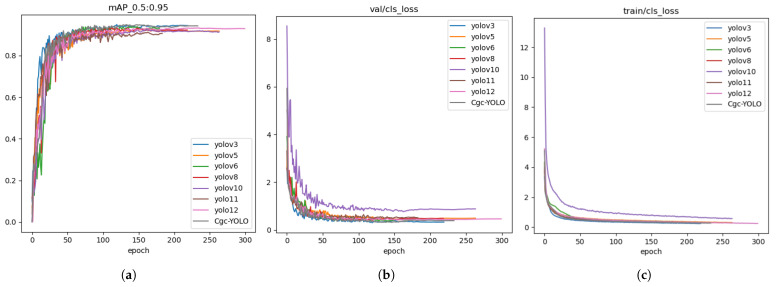
The performance of different models in the comparative experiment: (**a**) The mean average precision of the model within the IoU threshold range from 0.5 to 0.95; (**b**) the classification loss on the validation set. (**c**) The classification loss on the train set.

**Figure 10 sensors-25-05446-f010:**
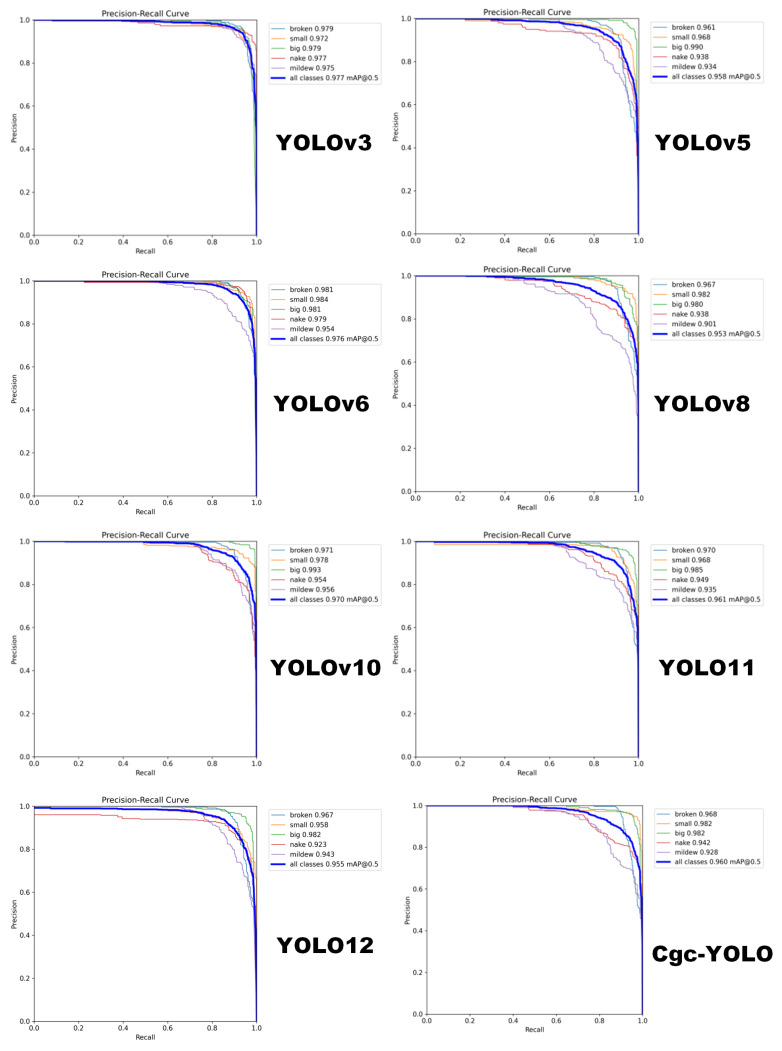
P–R curves of different models.

**Figure 11 sensors-25-05446-f011:**
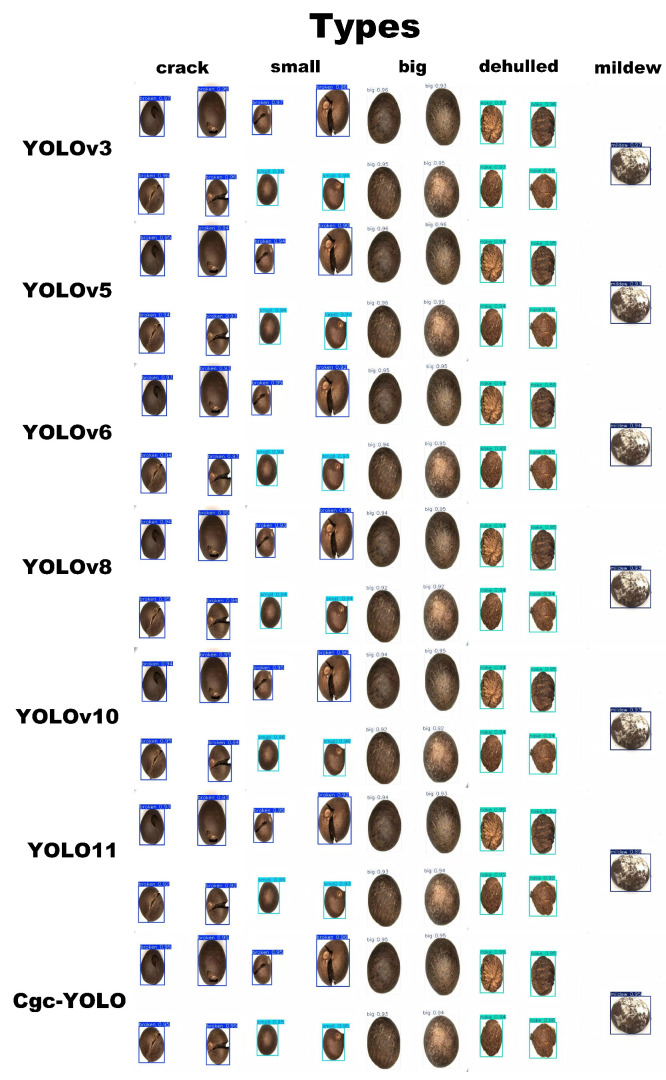
Example results of the visual comparison of the three defect types with the size of the good seed.

**Figure 12 sensors-25-05446-f012:**
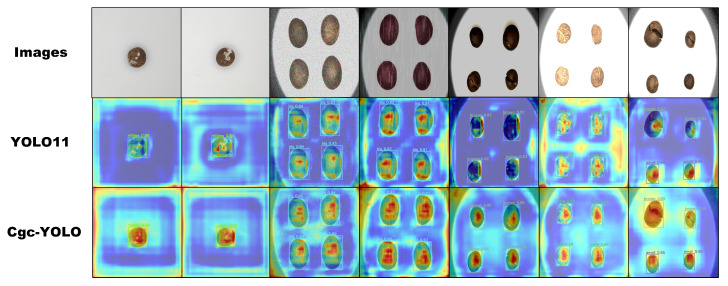
Grad-CAM visualization results of YOLO11 and Cgc-YOLO.

**Figure 13 sensors-25-05446-f013:**
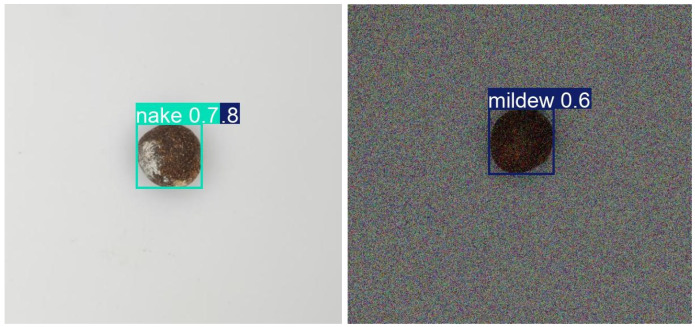
Example results of the false detection.

**Table 1 sensors-25-05446-t001:** Hyperparameter value of model in the experiment.

Hypreparameters	Values
Image size	640
Batch size	8
Epoch	300
Learning rate	0.01
Monmentum	0.937
Weight decay	0.0005

**Table 2 sensors-25-05446-t002:** Experimental configuration in the experiment.

Hardware/Software	Model/Version
GPU	NVIDIA GeForce RTX 4060 Laptop 8G
Computer System	Windows11
CUDA	12.7
Python	3.12
PyTorch	2.6.0
Visual Studio Code	1.99.2

**Table 3 sensors-25-05446-t003:** Results of the comparative experiment of CPCA module.

Mould	Box	R	mAP@50	mAP@50-95	GFLOPs	Param/M	F1-Score	FPS
YOLO11	0.899	0.882	0.953	0.918	6.3	2.58	0.886	414.78
YOLO11+CPCA-Neck	0.917	0.890	0.963	0.935	13.3	3.61	0.903	141.05
YOLO11+CPCA-Backbone	0.917	0.900	0.964	0.927	6.5	2.71	0.907	399.52
YOLO11+CPCA-B&N	0.917	0.900	0.963	0.935	13.3	3.62	0.903	139.21

**Table 4 sensors-25-05446-t004:** Results of the ablation experiment.

Mould	Box	R	mAP@50	mAP@50-95	Param/M	F1-Score	FPS
YOLO11	0.899±0.004	0.882±0.003	0.953±0.005	0.918±0.004	2.58	0.886±0.003	414.78±3.65
YOLO11+GhostBlock	0.919±0.003	0.912±0.002	0.975±0.004	0.954±0.003	3.14	0.913±0.003	352.70±2.10
YOLO11+CPCA-Neck	0.917±0.005	0.890±0.003	0.963±0.006	0.935±0.004	3.61	0.903±0.004	141.05±1.25
YOLO11+CPCA-Backbone	0.917±0.004	0.900±0.003	0.964±0.005	0.927±0.004	2.71	0.907±0.003	399.52±3.82
Cgc-YOLO	0.943±0.003	0.902±0.002	0.976±0.004	0.949±0.003	4.06	0.921±0.003	136.68±1.12

**Table 5 sensors-25-05446-t005:** Training results of different YOLO recognition models.

Mould	Box	R	mAP@50	mAP@50-95	Weight/MB	GFLOPs	Param/M	F1-Score
YOLOv3	0.924	0.939	0.977	0.950	197.4	261.8	98.45	0.930
YOLOv5	0.892	0.884	0.958	0.929	4.7	5.8	2.18	0.882
YOLOv6	0.908	0.912	0.970	0.939	8.6	11.5	4.15	0.906
YOLOv8	0.902	0.873	0.961	0.934	5.6	6.8	2.68	0.882
YOLOv10	0.911	0.894	0.960	0.926	5.8	6.5	2.26	0.900
YOLO11	0.899	0.882	0.953	0.918	5.5	6.3	2.58	0.886
YOLO12	0.902	0.881	0.954	0.932	5.2	5.8	2.50	0.885
RetinaNet	0.906	0.948	0.962	0.889	151.1	19.85	97.192	0.918
**Cgc-YOLO**	**0.943**	**0.902**	**0.976**	**0.949**	**8.5**	**14.1**	**4.06**	**0.921**

## Data Availability

Data are available upon request due to privacy or ethical restrictions. Data from this study are available from the corresponding authors upon request. Because of the privacy implications of the data in this study, these data are not publicly available.

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
