# Peer review of "Cgc-YOLO: A New Detection Model for Defect Detection of Tea Tree Seeds"

_sensors, 2025, doi:10.3390/s25175446_

Round 1

Reviewer 1 Report

Comments and Suggestions for Authors

This paper presents the Cgc-YOLO detection model, which incorporates GhostNetV2 as a lightweight framework within the YOLO network and the CPCA attention mechanism for feature integration. Both techniques are known to reduce computational complexity and improve model attention. Combining these methods into a single detection model is somewhat innovative in the context of specific tea seeds, but from an algorithmic perspective, it is rather an extension of existing approaches. Similar lightweight variants of YOLO have already been built for the detection of small objects. Cgc-YOLO extends this concept with an additional CPCA attention module, which is intended to improve the detection of complex defect features. However, it should be emphasized that the overall YOLO architecture remains unchanged, and the innovation comes down to the combination of known elements. The authors also developed a proprietary database of tea seed images with five classes, which can be considered a novel dataset in this field. Overall, however, the model achieves only a few percent improvement in mAP over the baseline YOLO (by approximately 2–3%), indicating a moderate performance gain. Compared to other object detection methods, Cgc-YOLO demonstrates its value primarily in the specific context of detecting defects in tea seeds. Although the applied improvements consist of previously known techniques, their combination and adaptation to this task represent a significant addition to the state of the art.

The problem of non-destructive detection of tea seed defects is important from the perspective of agricultural technology and seed storage. The proposed model, achieving very high accuracy (mAP@50 ~97.6%), indicates potential practical applications in automatic seed selection using image processing. Compared to the baseline YOLO11, the mAP increase of several percentage points is noticeable, although moderate. Cgc-YOLO also has the advantage of a small network size (approximately 8.5MB) and reduced computational load thanks to GhostNetV2, making it attractive for implementation on resource-constrained devices. Compared to the results of other works on seed or plant defect detection, the achieved results are very good. The scientific contribution can therefore be assessed as solid, primarily in the context of application: the paper contributes a dataset for further research and demonstrates that combining GhostNetV2 and CPCA in the YOLO network enables high precision. On the other hand, from the point of view of the general development of detection methods, the model does not introduce a groundbreaking architecture – rather, it tests known techniques on a new problem.

The research methodology has several shortcomings: the test sample comes from a single source (one tea variety, one photographic environment), which may limit the generalizability of the results. Furthermore, cross-validation results and tests on other sets are not provided, making it difficult to assess the model's stability. Furthermore, the use of the terms "YOLO11" and "YOLOv8–v11" in comparisons may introduce ambiguity; it is not widely known what "v10" or "v11" means in this context.

Comments on the Quality of English Language

The English in the article contains numerous stylistic and grammatical imperfections:
- in the abstract: the phrase "For coping with the above challenges" should read "To cope with the above challenges." The sentence is incomplete: "Comparative experimental evaluation shows that."
- the word "nudity" is used instead of what is likely "dehulled."
- "Albmentation" is used instead of the library name "Albumentations."
- the module names (such as C3k2, C2PSA) are not explained. In summary, the article requires substantial English improvement, both stylistic and grammatical, to meet the standards of scientific publications in English.

Author Response

Comment: The research methodology has several shortcomings: the test sample comes from a single source (one tea variety, one photographic environment), which may limit the generalizability of the results. Furthermore, cross-validation results and tests on other sets are not provided, making it difficult to assess the model's stability. Furthermore, the use of the terms "YOLO11" and "YOLOv8–v11" in comparisons may introduce ambiguity; it is not widely known what "v10" or "v11" means in this context.

The English in the article contains numerous stylistic and grammatical imperfections:
- in the abstract: the phrase "For coping with the above challenges" should read "To cope with the above challenges." The sentence is incomplete: "Comparative experimental evaluation shows that."
- the word "nudity" is used instead of what is likely "dehulled."
- "Albmentation" is used instead of the library name "Albumentations."
- the module names (such as C3k2, C2PSA) are not explained. In summary, the article requires substantial English improvement, both stylistic and grammatical, to meet the standards of scientific publications in English.

Response: Thank you very much for pointing this out. We have revised the paper according to your suggestions. We have explained the issue of the unity of the test samples in the Methods section. Because Longjing No. 43 was selected as the research material due to its wide cultivation, high genetic consistency and accessibility in both the Zhejiang and Jiangsu tea-producing regions. In addition, we have added a discussion section to illustrate the limitations of the results. Regarding the issue of model stability, we conducted multiple experiments in the ablation experiment and finally determined a confidence interval to evaluate the model stability. As for the research of Cgc-YOLO on other datasets, this will be our next plan. And thank you for raising the issue of ambiguous model names. We have conducted a thorough review and made corrections to the ambiguous parts. All "v11" has been corrected to "11". Thank you for your suggestions on the quality of English. The inappropriate phrases "For coping with the above challenges" and "Comparative experimental evaluation shows that." in the abstract have been revised. And the use of "dehulled" instead of "nudity", as well as the misspelled "Albmentation", has also been corrected. Explanations for "C3k2" and "C2PSA" have also been added. In addition, we have made extensive polishing of the English part of the paper. Thank you again for your suggestions and guidance to us.

Reviewer 2 Report

Comments and Suggestions for Authors

The article provides a detailed introduction to theconstruction of the model, experimental methods, results analysis, and a comparison with other traditional methods. Therefore, I recommend major revision this manuscript for publication. Below are my comments, which I hope will be helpful to the authors.

1Please provide a brief description of each of the upcoming sections of the paper in the introduction section. The introduction section is somewhat lengthy and includes some extraneous material. To enhance clarity and focus, it is advisable to provide a more detailed problem statement, which will better elucidate the motivation behind the current research.

2Some of the images could be more standardized and beautiful, such as Figures 1 and 2

3It might be a typo. I don't think the evaluation indicators are included in table 2

4Grad-CAM needs proper citations, and I recommend one or two articles. Just to mention a few:10.1016/j.foodcont.2024.110823;10.3390/s21175813

5To enhance the discussion, I suggest there should be a separate discussion section.

6Evaluation metrics are not enough, please add evaluation metrics.

7 Interpretability studies via SHAP value analysis or equivalent.

Author Response

Comment 1: Please provide a brief description of each of the upcoming sections of the paper in the introduction section. The introduction section is somewhat lengthy and includes some extraneous material. To enhance clarity and focus, it is advisable to provide a more detailed problem statement, which will better elucidate the motivation behind the current research.

Response: Thank you for your valuable feedback. We have revised the introduction section to more clearly clarify the problem solved by Cgc-YOLO and a brief description of each chapter was added. Specifically, our model has been enhanced with GhostBlock and CPCA on YOLO11 to better detect small and irregular defects, meeting the demands of non-destructive testing for recalcitrant seeds like tea tree seeds.

Comment 2: Some of the images could be more standardized and beautiful, such as Figures 1 and 2.

Response: Thank you for the suggestion. The contents of Figures 1 and 2 have been updated. Specifically, category annotations have been made in Figure 1 and the lighting conditions in Figure 2 have been modified.

Comment 3: It might be a typo. I don't think the evaluation indicators are included in table 2.

Response: We fully agree with your suggestion. This content has been deleted.

Comment 4: Grad-CAM needs proper citations, and I recommend one or two articles. Just to mention a few:10.1016/j.foodcont.2024.110823;10.3390/s21175813.

Response: Thank you for the suggestion. We agree that it is necessary for Grad-CAM to make some citations. Therefore, we summarize the contents of these two papers and add them in the relevant areas to make this part more logical.

Comment 5: To enhance the discussion, I suggest there should be a separate discussion section.

Response: Thank you for your valuable feedback. We conducted a separate discussion section before the conclusion chapter, which included discussions on Result Interpretation and Limitation. These discussions made the article more comprehensive.

Comment 6: Evaluation metrics are not enough, please add evaluation metrics.

Response: We fully agree with your suggestion. Based on the original ones, we have added new evaluation indicators, such as F1-score and FPS. And add it to the comparison of various models to enhance the multi-faceted consideration of the advantages and disadvantages of the models.

Comment 7: Interpretability studies via SHAP value analysis or equivalent.

Response: We agree with the importance of interpretability study. Therefore, we used Grad-CAM for the full feature analysis to demonstrate that Cgc-YOLO has a significant improvement in attention in different features and pays more attention to the target area.

Reviewer 3 Report

Comments and Suggestions for Authors

1. The YOLO series of models has reached YOLOv13. It is recommended to add comparative experiments for models after YOLOv11.  
2. It is recommended not to compare only YOLO models but to include comparative experiments with other foundational object detection models, such as Faster R-CNN, etc.
3. The text in Figures 8 and 9 is too large, while the experimental results are too small. It is recommended to enlarge the scale of the experimental results or adopt a more visually appealing presentation format.  
4. The color scheme in Figure 5 is too monotonous. It is recommended to add some accent colors to enhance readability.  
5. In Section 2.4.1, it is recommended to provide a detailed description of the improved Cgc-YOLO model and the specific enhancements made. The original YOLOv11 model diagram can be omitted.
6. In Table 5, it is recommended to change “ours” to “Cgc-YOLO.”
7. It is recommended to clarify what (a) and (b) represent in the description of the model architecture in Figure 5.
8. It is recommended to rename “yolo11-Ghost-CPCA” in Figure 7 to the model's own name.

Author Response

Comment 1: The YOLO series of models has reached YOLOv13. It is recommended to add comparative experiments for models after YOLOv11.  

Response: Thank you for your valuable feedback. For other models in the YOLO series, we have newly added YOLO12 for our comparative experiments, and we have also introduced new evaluation metrics to make the comparative experiments more comprehensive.

Comment 2: It is recommended not to compare only YOLO models but to include comparative experiments with other foundational object detection models, such as Faster R-CNN, etc.

Response: Thank you for the suggestion. For models other than the YOLO series, we selected the RetinaNet model for our comparison model, where the dataset was transformed from the dataset of this experiment. Make the comparative experiments more comprehensive and reliable.

Comment 3: The text in Figures 8 and 9 is too large, while the experimental results are too small. It is recommended to enlarge the scale of the experimental results or adopt a more visually appealing presentation format.  

Response: We fully agree with your suggestion.We have updated the presentation of Figure 8 to make it more visible. For Figure 9, we have expanded the scale of the results to a certain extent to make them clearer.

Comment 4: The color scheme in Figure 5 is too monotonous. It is recommended to add some accent colors to enhance readability.  

Response: Thank you for the suggestion. For the problem of monotonous colors in Figure 5, we have solved it. The same color is used in the same module, and different colors are adopted to represent the process, making Figure 5 more beautiful.

Comment 5: In Section 2.4.1, it is recommended to provide a detailed description of the improved Cgc-YOLO model and the specific enhancements made. The original YOLOv11 model diagram can be omitted.

Response: Thank you for your valuable feedback. It is indeed necessary to explain the Cgc-YOLO model in Section 2.4.1. Therefore, we have added a detailed description of the Cgc-YOLO enhancements in Section 2.4.1 to make it more complete.

Comment 6: In Table 5, it is recommended to change “ours” to “Cgc-YOLO.”

Response: We fully agree with your suggestion. We have now adjusted it back to "Cgc-YOLO" and added new evaluation metrics in the table.

Comment 7: It is recommended to clarify what (a) and (b) represent in the description of the model architecture in Figure 5.

Response: We fully agree with your suggestion. We have provided the corresponding explanation below this figure. (a) in Figure 5 is Architecture of the Ghost_Block module; (b) is Architecture of the Ghost_HGBlock module.

Comment 8: It is recommended to rename “yolo11-Ghost-CPCA” in Figure 7 to the model's own name.

Response: We agree with your suggestion. The name in this figure has been re-modified to "Cgc-YOLO", and in this figure, we have newly added YOLO12 to improve our content.

Reviewer 4 Report

Comments and Suggestions for Authors

The manuscript presents a lightweight and accurate YOLO-based model for detecting defects in tea tree seeds. However, it has significant issues with clarity, grammar, organization, and reproducibility. The methodology is solid, but the writing needs extensive improvement. Details on statistical validation and reproducibility are absent, and terms like “YOLO11” require clarification. A major revision is necessary to address these issues. Below are my comments:

  1. The abstract summarizes the motivation, model architecture, and performance of Cgc-YOLO, reporting high accuracy. However, it lacks organization and precision, introducing “YOLO11” without context and containing incomplete phrases (e.g., “Comparative experimental evaluation shows that”). It also dives too deeply into technical details without sufficient explanation and has grammatical issues. The abstract should be structured to clearly outline the problem, methodology, and key results, while explaining technical components like GhostBlock and CPCA.
  2. The introduction highlights the importance of seed quality monitoring and the limitations of traditional methods, tracing the evolution from manual inspection to machine learning and deep learning. Yet, it suffers from excessive citations, weak transitions, and inadequate synthesis of previous work. The mention of Swin Transformers lacks meaningful comparison with CNN-based YOLO models, and “YOLO11” is not explained. The introduction should be restructured: begin with the problem background, review existing techniques in a logical progression, identify gaps in current YOLO approaches for seed defect detection, and clearly state the novelty of Cgc-YOLO.
  3. The methodology section is detailed but overly verbose. In the seed preparation subsection, it reads more like lab instructions, focusing on temperatures and humidity without justifying the choice of Longjing No. 43. The image acquisition subsection describes hardware in excessive detail, listing brand names and specific configurations that could be summarized. The image processing subsection effectively uses the Albumentations library but lacks explanations for parameter choices, such as the relevance of noise variance and hue adjustments.
  4. The core model architecture section covers YOLO11, GhostNetV2, and CPCA, introducing improvements like the Ghost_Block and CPCA in the neck network. However, it lacks clarity on what YOLO11 is, with equations presented without context. The CPCA attention mechanism is complex and requires more precise explanations and flow diagrams. This section should be reorganized with clearer headings and reduced mathematical complexity.
  5. The results and discussion section is missing essential details, such as dataset class distribution, image counts, and convergence metrics. Definitions for performance metrics are too textbook-like and could be moved to an appendix. While the CPCA module comparison is informative, the phrasing is awkward and lacks visual representation. The ablation studies effectively show contributions to performance, but results are not reported with confidence intervals, which obscures the significance.
  6. In comparative experiments, Cgc-YOLO is evaluated against YOLOv3 but lacks clarity on whether all YOLO models were retrained or used pre-trained weights, raising fairness concerns. Claims of performance gains are made without statistical validation or an analysis of failure cases. Visualisation results, including detection outputs and Grad-CAM heatmaps, are helpful but could include quantitative metrics and more detailed explanations.
  7. The conclusion is generic and repetitive. It doesn't acknowledge major limitations. Future directions should be more specific.
  8. There’s no citation of works like YOLO-Nano, MobileNet-SSD, Tiny-YOLOv4, which are directly relevant to the lightweight objective of Cgc-YOLO.
Comments on the Quality of English Language

The manuscript requires a major language revision. The writing lacks fluency, academic tone, and clarity in conveying technical content. It is recommended that the authors rewrite each section with careful attention to grammar, syntax, scientific diction, and structural coherence. 

Author Response

Comment 1: The abstract summarizes the motivation, model architecture, and performance of Cgc-YOLO, reporting high accuracy. However, it lacks organization and precision, introducing “YOLO11” without context and containing incomplete phrases (e.g., “Comparative experimental evaluation shows that”). It also dives too deeply into technical details without sufficient explanation and has grammatical issues. The abstract should be structured to clearly outline the problem, methodology, and key results, while explaining technical components like GhostBlock and CPCA.

Response: Thank you for your valuable feedback. Your suggestions have greatly enhanced the readability of the abstract. We have rewritten the abstract, describing the problems, methods and key results of this study, and explained the GhostBlock and CPCA technology components as per your suggestions.

Comment 2: The introduction highlights the importance of seed quality monitoring and the limitations of traditional methods, tracing the evolution from manual inspection to machine learning and deep learning. Yet, it suffers from excessive citations, weak transitions, and inadequate synthesis of previous work. The mention of Swin Transformers lacks meaningful comparison with CNN-based YOLO models, and “YOLO11” is not explained. The introduction should be restructured: begin with the problem background, review existing techniques in a logical progression, identify gaps in current YOLO approaches for seed defect detection, and clearly state the novelty of Cgc-YOLO.

Response: Thank you for the suggestion. Your suggestions will enhance the logic of the introduction section. The introduction part has been reorganized under your suggestions, including: introducing the existing technology from a logical background after raising the question, identifying the gap in YOLO seed defect detection, and introducing Cgc-YOLO. Meanwhile, a deeper comparison was made between Swim Transformer and CNN, and the citations were streamlined, with some citation contents deleted.

Comment 3: The methodology section is detailed but overly verbose. In the seed preparation subsection, it reads more like lab instructions, focusing on temperatures and humidity without justifying the choice of Longjing No. 43. The image acquisition subsection describes hardware in excessive detail, listing brand names and specific configurations that could be summarized. The image processing subsection effectively uses the Albumentations library but lacks explanations for parameter choices, such as the relevance of noise variance and hue adjustments.

Response: We fully agree with your suggestion.Your suggestions make our work more reasonable. In the method, we discussed the selection of Longjing 43 and explained why it was reasonable and valuable to choose it as the experimental material. And we have streamlined the overly detailed hardware description. We also explained the parameter selection in the Albumentations library one by one, such as noise, raindrops, and so on.

Comment 4: The core model architecture section covers YOLO11, GhostNetV2, and CPCA, introducing improvements like the Ghost_Block and CPCA in the neck network. However, it lacks clarity on what YOLO11 is, with equations presented without context. The CPCA attention mechanism is complex and requires more precise explanations and flow diagrams. This section should be reorganized with clearer headings and reduced mathematical complexity.

Response: Thank you for the suggestion. Your suggestions have enhanced our logic. In the core framework, we have newly added an introduction to YOLO11, connecting the contextual content. And the CPCA section has been reorganized. For the content of CPCA,  replace the title with "Channel Prior Convolutional Attention". In addition, we have carried out a major optimization of the flowchart and added an explanation section to its content.

Comment 5: The results and discussion section is missing essential details, such as dataset class distribution, image counts, and convergence metrics. Definitions for performance metrics are too textbook-like and could be moved to an appendix. While the CPCA module comparison is informative, the phrasing is awkward and lacks visual representation. The ablation studies effectively show contributions to performance, but results are not reported with confidence intervals, which obscures the significance.

Response: Thank you for your valuable feedback. Your suggestions have made our work more perfect. Our discussion of the results does indeed lack detailed information about the dataset. We integrated it and added it to the article, but because it is related to the dataset, we added it to the methods section. Regarding the content of the CPCA comparison module, we have newly added the comparison curve graph of this comparison experiment, making the results more visually expressive. In the ablation experiment, we conducted multiple experiments and determined a confidence interval for the relevant parameters, which we represented.

Comment 6: In comparative experiments, Cgc-YOLO is evaluated against YOLOv3 but lacks clarity on whether all YOLO models were retrained or used pre-trained weights, raising fairness concerns. Claims of performance gains are made without statistical validation or an analysis of failure cases. Visualisation results, including detection outputs and Grad-CAM heatmaps, are helpful but could include quantitative metrics and more detailed explanations.

Response: We fully agree with your suggestion. In the comparative experiments, all YOLO models were retrained instead of using preweights, which ensured fairness among the model comparisons. Regarding the definition of performance indicators, we have streamlined it to make it more readable In addition, we conducted t-tests on Cgc-YOLO and other models. Except for YOLOv3, the results of all others were less than 0.005, making up for the previous deficiencies in statistical results. For the quantitative indicators and explanations of Grad-CAM, we not only analyzed all its features, but also added new explanatory content to the work of Grad-CAM, comparing and explaining it with the previous table data, making our work clearer.

Comment 7: The conclusion is generic and repetitive. It doesn't acknowledge major limitations. Future directions should be more specific.

Response: We fully agree with your suggestion. Your viewpoint will make our work clearer. The conclusion section provides a more specific description of the future direction, and we have newly added a discussion section, which contains the description of the Limitation in this work.

Comment 8: There’s no citation of works like YOLO-Nano, MobileNet-SSD, Tiny-YOLOv4, which are directly relevant to the lightweight objective of Cgc-YOLO.

Response: We agree with your suggestion. We agree that references to lightweighting are highly necessary. Models such as YOLO-Nano, MobileNet-SSD, and Tiny-YOLOv4 have been cited in the introduction section, and the goal of lightweighting has been discussed in separate paragraphs.

Round 2

Reviewer 2 Report

Comments and Suggestions for Authors

All issues have been addressed and I recommend acceptance of this paper

Author Response

We sincerely thank you for the positive feedback and recommendation for acceptance. Your constructive comments during the review process have been invaluable in improving the quality and clarity of our work. We are truly grateful for your time and effort.

Reviewer 4 Report

Comments and Suggestions for Authors

The authors have made some revisions; however, most changes do not resolve the deeper weaknesses in novelty, methodological rigor, statistical reporting, and critical discussion. Many of your original comments remain only partially addressed. The manuscript still: (i) Overstates novelty and impact; (ii) Lacks rigorous statistical validation; (iii) Has poor integration of figures/tables with the narrative; (iv) Provides insufficient detail on training fairness and experimental reproducibility; (v) Avoids a serious exploration of limitations or alternative explanations. I would still recommend Major Revisions before considering acceptance.

While some citations have been removed in the Introduction, the narrative is still heavy without proper synthesis. The comparison between Swin Transformer and CNN-based YOLO models is limited to vague statements rather than a critical analysis of architectural trade-offs. “YOLO11” is now defined, but the description is buried rather than clearly introduced early. The novelty claim of Cgc-YOLO is still weak - lightweight YOLO variants are now cited but not discussed in detail, making the “gap” appear overstated. Overall, the section still fails to frame the necessity and uniqueness of this work convincingly.

While the authors trimmed some hardware details, the methods section remains overly verbose in areas of low value (lab conditions). Still, it lacks depth in justifying choices, e.g., why Longjing No. 43 was chosen over other varieties beyond generic “reasonable and valuable” phrasing. Albumentations parameter explanations are present but minimal, lacking justification for their relevance to the real-world variability of seed images. The LCA-style detail in preprocessing is not clearly linked to its impact on model performance. Some subsections remain better suited for supplementary material rather than the main text.

The addition of a YOLO11 introduction is excellent, but not smoothly integrated. The CPCA section is reorganized and includes a revised flowchart, yet the mathematical presentation is still poorly contextualized and lacks an intuitive explanation. Equations are provided without discussing assumptions or computational costs, which matters for a paper claiming “lightweight” efficiency. The figure is improved, but still not self-explanatory; s readers must cross-reference the text to understand it.

In results, some of your original requests are partially met. DDataset class distribution and image counts are now present, but they are placed in the methods rather than the results, which disrupts the flow. Convergence curves are still missing, although they are essential for training analysis. The CPCA comparison now has a visual curve, but the caption and description remain vague, failing to link it to the claimed benefits. The ablation study now includes confidence intervals, but statistical reporting is inconsistent as intervals are sometimes missing, and there’s no explanation of how they were computed. The “discussion” element is still weak, mostly restating results instead of interpreting their significance, limitations, and broader impact.

The authors claim retraining of all YOLO models for fairness, but details of training hyperparameters, dataset splits, and hardware parity are insufficient. T-tests are now mentioned, but the reporting is shallow as no effect sizes or multiple comparison corrections are applied. Failure cases are still absent, and qualitative detection results are cherry-picked to highlight successes. Grad-CAM explanations are expanded but still do not address the meaning of the highlighted regions in terms of feature learning or decision errors.

The conclusion is slightly improved, with a limitations paragraph added, but it reads as a token acknowledgment rather than a critical reflection. Limitations are described in generic terms and do not address core issues like dataset representativeness, potential overfitting, or lack of external validation. Future work suggestions were not directly tied to the shortcomings identified in the study.

Lightweight detection references are added, but still in a cursory manner. Several citations are not directly integrated into the discussion - they appear to have been added primarily to appease reviewer comments rather than to strengthen the narrative.

Comments on the Quality of English Language

The manuscript needs comprehensive language polishing with a focus on grammar, sentence simplification, removal of redundancies, and improved cohesion between sections.

Author Response

Comment 1: While some citations have been removed in the Introduction, the narrative is still heavy without proper synthesis. The comparison between Swin Transformer and CNN-based YOLO models is limited to vague statements rather than a critical analysis of architectural trade-offs. “YOLO11” is now defined, but the description is buried rather than clearly introduced early. The novelty claim of Cgc-YOLO is still weak - lightweight YOLO variants are now cited but not discussed in detail, making the “gap” appear overstated. Overall, the section still fails to frame the necessity and uniqueness of this work convincingly.

Response: Thank you for your valuable suggestions. In response to your suggestions, we have made changes to the introduction part, placed the introduction of YOLO11 at the front position, and conducted a more in-depth comparison and analysis between Swin Transformer and the CNN-based YOLO model. In addition, the lightweight YOLO section has been restructured and targeted discussions have been added. Thank you again for your suggestions, which have made our work presentation more reasonable and necessary.

Comment 2: While the authors trimmed some hardware details, the methods section remains overly verbose in areas of low value (lab conditions). Still, it lacks depth in justifying choices, e.g., why Longjing No. 43 was chosen over other varieties beyond generic “reasonable and valuable” phrasing. Albumentations parameter explanations are present but minimal, lacking justification for their relevance to the real-world variability of seed images. The LCA-style detail in preprocessing is not clearly linked to its impact on model performance. Some subsections remain better suited for supplementary material rather than the main text.

Response: Thank you for the suggestion. We have streamlined the method section. We have made some deletions to the laboratory conditions and other contents mentioned. and clarified the reasons for choosing Longjing 43 :Not only for its wide cultivation and genetic uniformity but also because it is a high-value cultivar in the Chinese tea industry and fine-grained surface defects—such as cracks, bare areas, and mold growth, which visually subtle and challenging to detect. These characteristics make it an ideal benchmark for evaluating small-defect detection models. At the same time, on the original basis, we have provided more explanations for the Albumentations parameter and added the actual situations in the real world corresponding to each enhancement step. Finally, we discuss and analyze the impact of enhancement processing on model performance.

Comment 3: The addition of a YOLO11 introduction is excellent, but not smoothly integrated. The CPCA section is reorganized and includes a revised flowchart, yet the mathematical presentation is still poorly contextualized and lacks an intuitive explanation. Equations are provided without discussing assumptions or computational costs, which matters for a paper claiming “lightweight” efficiency. The figure is improved, but still not self-explanatory; s readers must cross-reference the text to understand it.

Response: Thank you for your valuable opinions on our work. This will enhance the logic and readability of this article. For the YOLO11 section, some adjustments have been made to make its introduction smoother and more reasonable. For the CPCA section, we reorganized it and added contextual explanations. First, the design goals of the module were expounded. Then, the architecture of CPCA was visually presented in combination with a flowchart, and an explanatory part was added to the diagram. Subsequently, the specific process of CPCA was described, and the mathematical process of this process was summarized into a formula. Afterwards, we compared and discussed the computing costs of CPCA and CBAM. Finally, summarize this part and present the final output feature formula at the end. Make it easier for readers to understand our work.

Comment 4: In results, some of your original requests are partially met. DDataset class distribution and image counts are now present, but they are placed in the methods rather than the results, which disrupts the flow. Convergence curves are still missing, although they are essential for training analysis. The CPCA comparison now has a visual curve, but the caption and description remain vague, failing to link it to the claimed benefits. The ablation study now includes confidence intervals, but statistical reporting is inconsistent as intervals are sometimes missing, and there’s no explanation of how they were computed. The “discussion” element is still weak, mostly restating results instead of interpreting their significance, limitations, and broader impact.

Response: Thank you for the suggestion. Your suggestions have enhanced our logic. Based on your suggestion, we have now placed the Dataset class distribution and image counting in the methods section. The convergence curves were placed in the comparative experiments, including val/cls_loss and train/cls_loss, where the convergence situations of each model were included, such as YOLO11, Cgc-YOLO, etc. The title of the CPCA comparison has been modified, and the description and discussion sections have also been revised. It clarifies the comparison of CPCA at different positions and the benefits it brings, and explains the reasons for ultimately choosing CPCA-Neck. In the ablation experiment, for the calculation of the confidence interval, we used the formula of the 95% confidence interval for calculation and summarized it in the ablation experiment section. The confidence interval loss problem mentioned should be the Param of the corresponding model, as the number of parameters is determined by the network design. Therefore, the Param of each model is always fixed and has nothing to do with training fluctuations. In the discussion section, we have made modifications and additions to the content, paying more attention to its interpretation in terms of meaning and acknowledging our limitations. Besides, a Broader Impact section has been added to discuss the extensive influence.

Comment 5: The authors claim retraining of all YOLO models for fairness, but details of training hyperparameters, dataset splits, and hardware parity are insufficient. T-tests are now mentioned, but the reporting is shallow as no effect sizes or multiple comparison corrections are applied. Failure cases are still absent, and qualitative detection results are cherry-picked to highlight successes. Grad-CAM explanations are expanded but still do not address the meaning of the highlighted regions in terms of feature learning or decision errors.

Response: Thank you for your valuable feedback. Your suggestions have made our work more perfect. In the comparative experiments, all YOLO models were conducted under the same experimental environment and parameters. We have listed all the relevant detailed experimental information in the comparative experiment module. For the t-test, we now add the effect sizes and Bonferroni correction to make our statistical analysis more convincing. For the failure cases, we placed them under the limitations in the discussion section, discussed the specific causes of misjudgment, and presented the failure cases using pictures. In Grad-CAM, we combine two models to expound the meanings of highlighting regions in terms of learning and misjudgment: Cgc-YOLO is more concentrated on defect regions, indicating that it has learned to focus on encoding the relevant features of defects after improvement; However, YOLO11 has a weak response in some defect areas and is easily disturbed by noise, leading to incorrect judgments.

Comment 6: The conclusion is slightly improved, with a limitations paragraph added, but it reads as a token acknowledgment rather than a critical reflection. Limitations are described in generic terms and do not address core issues like dataset representativeness, potential overfitting, or lack of external validation. Future work suggestions were not directly tied to the shortcomings identified in the study.

Response: We fully agree with your suggestion. And we made extensive revisions to the restrictive paragraphs of the conclusion and discussed the challenges of external validation and detection of minor defects in the dataset. And in response to the existing problems mentioned above, we have made plans for future work, making our work more comprehensive and targeted

Comment 7: Lightweight detection references are added, but still in a cursory manner. Several citations are not directly integrated into the discussion - they appear to have been added primarily to appease reviewer comments rather than to strengthen the narrative.

Response: We fully agree with your suggestion. Your viewpoint will make our work complete and logical. We place lightweight detection in the introduction section and conduct targeted analysis on it. Not only does it enhance the narrative, but it also naturally leads to the necessity of improving YOLO. It has now become an indispensable part of the introduction

Comment 8: The manuscript needs comprehensive language polishing with a focus on grammar, sentence simplification, removal of redundancies, and improved cohesion between sections.

Response: We agree with your suggestion. Following your suggestion, we have made extensive polishing of the English part of the paper. Thank you again for your suggestions and guidance to us.
